# Plant Spacing Effects on Stem Development and Secondary Growth in *Nicotiana tabacum*

**Na Xu** [1,†], **Lin Meng** [1,†], **Fang Tang** [2], **Shasha Du** [1], **Yanli Xu** [1], **Shuai Kuang** [1], **Yuanda Lv** [3], **Wenjing Song** [1], **Yang Li** [1], **Weicong Qi** [3,\*] and **Yu Zhang** [1,\*]

1    Key Laboratory of Tobacco Biology and Processing, Ministry of Agriculture, Tobacco Research Institute, Chinese Academy of Agricultural Sciences (CAAS), Qingdao 266101, China; xuna@caas.cn (N.X.); menglin@caas.cn (L.M.); wdsh0405@163.com (S.D.); xu87147519@163.com (Y.X.); kabashuai@sina.com (S.K.); songwenjing@caas.cn (W.S.); liyang926400@163.com (Y.L.)
2    State Key Laboratory of Tree Genetics and Breeding, Key Laboratory of Tree Breeding and Cultivation of the National Forestry and Grassland Administration, Research Institute of Forestry, Chinese Academy of Forestry, Beijing 100091, China; tangfangcaf@126.com
3    Excellence and Innovation Center, Jiangsu Academy of Agricultural Sciences (JAAS), Nanjing 210014, China; lyd0527@126.com
\*    Correspondence: weicong_qi@163.com (W.Q.); zhangyu02@caas.cn (Y.Z.)
†    These authors contributed equally to this work.

**Abstract:** Plant spacing usually refers to distances between plants within and between rows in the field. Different spacing in crop planting would generally influence the size, plant architecture, economic productivity, etc. The present research provided a time course monitoring of stem development in tobacco with different plant spacing. The result showed that cambium activity, vascular bundle thickness, lignin, cellulose, and hemicellulose content, as well as the macronutrient deposition in the stem varied because of the different plant spacing. Furthermore, the genes (*NtHB8s* and *NtNST3s*) coding the homologs of HB8 and NST3 transcription factors, which are involved in plant secondary growth, were cloned in tobacco. In the time course, they also indicated diverse expression patterns among altered plant-spacing treatments. Their transcriptomic activities were validated, and the motifs that might bind transcription factors in their promoter regions were predicted. Promoters of *NtHB8s* and *NtNST3s* genes were rich in light-response elements; as a result, light might be the main environmental factor in plant spacing to regulate stem secondary growth.

**Keywords:** stem development; secondary growth; plant spacing; secondary cell wall; gene expression; cis-acting element





## 1. Introduction

Stem is a central part of the plant that supports the body, connects various body parts, and transports important substances [1,2]. The stem development in crops can influence several agronomic traits, which are related to the cultivation and production of the crop. These traits include plant stand, lodging resistance, and crop yield. Therefore, stem development is an important factor in crop agronomy. Adequate stem development ensures proper plant stand, lodging resistance, and crop yield. The growth of a stem can be divided into two stages: primary growth and secondary growth. Primary growth in plants refers to the growth in length or height of the plant body. It occurs at the apical meristems, which are regions of actively dividing cells located at the tips of stems and roots. Secondary growth is different from primary growth, which is responsible for the increase in the length of the stem and roots. Secondary growth occurs in two main regions: the vascular cambium and the cork cambium [3].

Secondary growth results in thickening: at this stage, asymmetric divisions of cambium give rise to secondary phloem outward and secondary xylem inward [4,5]. Arabidopsis has always been used as a model plant to research stem development [1,4,6–9]. There is

also research that compares major regulators of stem secondary growth between poplar and Arabidopsis [10]. Phloem and xylem, which are divided from cambium, are the main components of vascular bundles in plants [11]. To achieve their roles as conduits of water and nutrients, as well as to support their whole body [12], the cells in plant vascular tissues show high differentiation, and their cell walls undergo highly specialized modifications [2].

The processes of cell differentiation in xylem and phloem tissues and secondary cell wall formation have been intensively studied, and many regulatory genes have been identified and characterized [13]. For instance, *ATHB-8* in Arabidopsis, a member of the HD-ZIPIII family, regulate by auxin positively, an early marker of the procambial cells and the cambium during vascular regeneration after wounding [14,15]. The overexpression of *ATHB-8* increases the number of xylem cells in the vascular bundle of the stem [16]. Another Arabidopsis gene *AtNST3* plays a crucial role in the formation of secondary walls in woody tissues. Overexpression of AtNST3 induced ectopic secondary wall thickenings in various aboveground tissues [17,18]. In poplar, *PtrHB7* and *PtrHB8* are close homologs of Arabidopsis *ATHB-8* [19,20], and suppression of *PtrHB7/PtrHB8* expression impedes secondary xylem differentiation [19]. *PtrWND1A/ PtrWND1B*, which are homologs of Arabidopsis *AtNST3*, are related to secondary cell wall formation in poplar [21,22].

Crop spacing can directly affect the stem development of the crop. The distance between the plants affects the availability of light, water, and nutrients that each plant receives, which can affect stem growth. Moreover, crop yield, plant morphological structure, and disease resistance are closely related to plant spacing [23–27]. The effect of plant spacing on anatomical structure, chemical composition, and nutrient transport of plant stem has not been studied in detail to date.

In the present research, we demonstrated an experiment in tobacco field cultivation with varied plant-spacing treatments. Stem development and the anatomical structure in it were monitored in a dynamic manner. The result showed different plant spacing effects on cambium cells and their second growth, which led to thicker phloem and xylem in the stem. Furthermore, the varied plant-spacing treatments influenced the expression of the crucial genes (*NtHB8s* and *NtNST3s*) regulating stem second growth, as well as the nutrient distribution in the plant.

## 2. Materials and Methods

### 2.1. Plant Material and Growth Conditions

The experiment was carried out in Huili City, Sichuan Province, China, ($27°12'$ N, $101°52'$ E) in the spring 2022. *Nicotiana tabacum* cultivar ZC208, which adapts to the local climate, was selected. Seeds were sown on the seedling substrate and grown in a greenhouse. After the seedling stage (Figure 1), tobacco seedlings were moved to the field. Four different plant-spacing treatments were set, which were 30, 40, 50, and 60 cm. The row space of different treatments was the same, which was 120 cm. This experiment was conducted by employing a randomized block design, and each treatment had three repeats. Compound fertilizer ($m(N):m(P_2O_5):m(K_2O) = 1:1:2.5$) was applied to the field, with a total nitrogen application rate of 67.5 kg·ha$^{-1}$.

At the rosette stage (Figure 1), when the stem pitch of nodes between the 4th and 5th leaf positions (counted from shoot to root) reached 1 cm, the node between the 4th and 5th leaf positions was marked with a plastic rope. The stem circumference of the marked nodes was measured every 4 days. Samples used for stem girth, stem anatomic structure, physiological index, and gene expression analysis were sampled 0, 4, 12, and 40 days after marking.

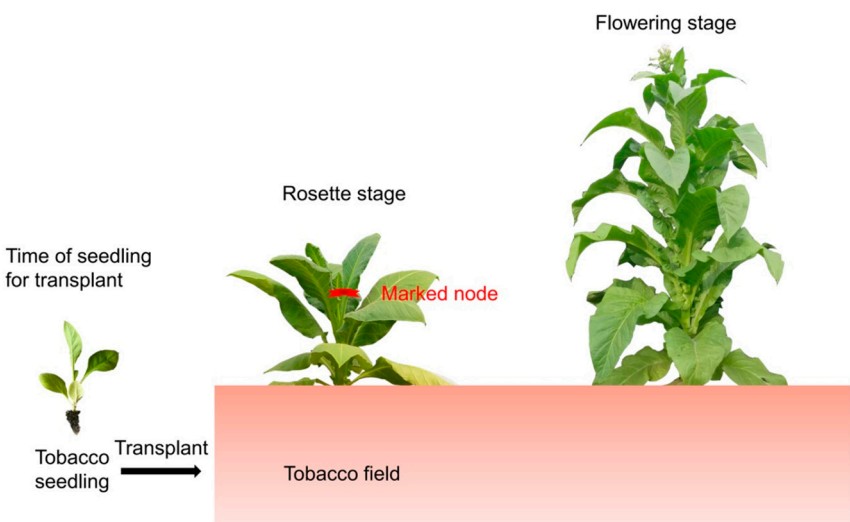

**Figure 1.** Tobacco development period. After germination the seedlings were cultivation in pot for a month, and then they were transplanted into the filed. Stem nodes were marked at the rosette stage (the number of leaves was 12–13). The flowering stage was the stage of the first central flower opening.

### 2.2. Measurements of Stem Girth and Microscopy

Because the stem of tobacco was an irregular circle, the soft ruler was used to measure stem girth. Samples for cross section observation were stocked in FAA (70% ethanol, 5% formaldehyde, 5% acetic acid) and were sectioned using a vibration slicer (Leica VT1000 S Vibratome, Leica Microsystems, Wetzlar, Germany) to a thickness of 25 μm. Sections were stained with Toluidine blue (T3260, Sigma Aldrich, St. Louis, MO, USA) and observed under the light microscope (DMC2900, Leica Microsystems, Wetzlar, Germany) [28,29]. The anatomical structure of the tobacco stem cross section is shown in Figure 2.

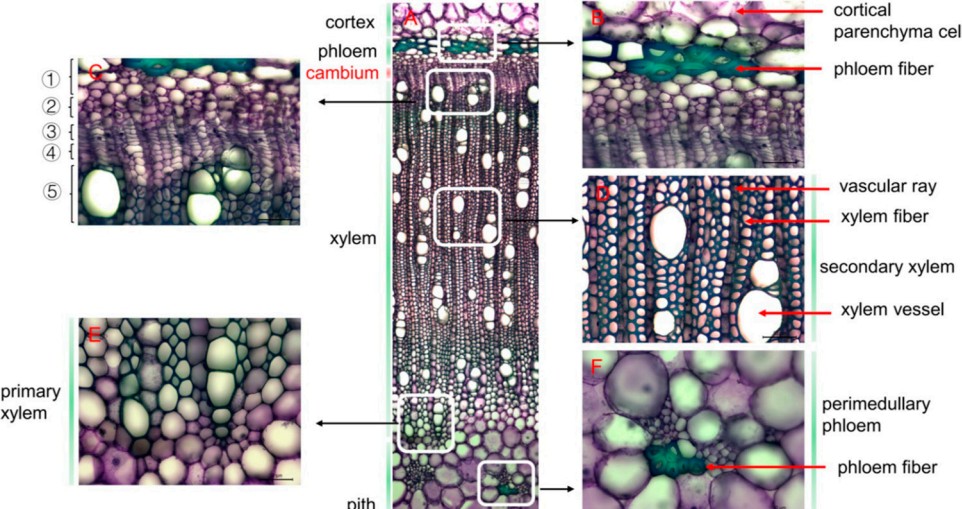

**Figure 2.** (**A**) Anatomical structure of the tobacco stem cross section mainly includes the cortex, phloem, cambial, xylem, and pith. Because the exact developmental stages were still undetermined, the bars describing particular developmental zones are approximate. (**B**) Detailed illustration of phloem. (**C**) Detailed illustration of cambium. ① mature phloem, ② phloem differentiation, ③ cambial cell divisions, ④ xylem differentiation, ⑤ secondary cell wall formation. (**D**) Detailed illustration of secondary xylem. (**E**) Detailed illustration of primary xylem. (**F**) Detailed illustration of perimedullary phloem. The scale bar of (**A**) is 100 μm, and the scale bar of (**B**–**E**) is 50 μm.

### 2.3. Chemical Composition of Secondary Cell Wall and Nutrient Content Analysis

At the flowering stage (Figure 1), the stems and leaves of the marked position were sampled. Samples were heat-treated at 105 °C for 30 min and dried at 65 °C to a constant weight and weighed for secondary cell wall and nutrient content analysis.

The contents of lignin, cellulose, and hemicellulose in the samples were determined by weighing [30,31]. Empty filter bags were dried in the oven and heat-treated at 105 °C for 4 h and then weighed as M1. A total of 0.5 g of sample (passing 20 mesh sieve) was placed into the filter bags and weighed as M2. The filter bags were boiled in neutral detergent (5 g/L anhydrous sodium sulfite, 30 g/L sodium dodecyl sulfate solution, 18.61 g/L disodium EDTA, 6.81 g/L sodium tetraborate, 30 g/L sodium dodecyl sulfate, 10 mL/L ethylene glycol ether) for 1 h. Then, the filter bags were taken out, washed with hot water, soaked in acetone for 1–2 h, and washed with acetone until the liquid was colorless. The filter bags were put in the fume hood to volatilize acetone for 1–2 h, placed into the oven at 105 °C for 4 h, and weighed as M3. Next, the filter bags were put into acid detergent (20 g/L cetyltrimethylammonium bromide), boiled repeatedly for 1 h, washed with hot water, washed with acetone until acetone volatilized, and then dried and weighed as M4. The dried filter bags were put into 72% sulfuric acid for 3 h and washed with water until they were free of acid. Then, the filter bags were soaked in acetone for 1–2 h, and the acetone washing, acetone volatilization, and filter bag drying steps were repeated. The weight of the dried filter bags was recorded as M5. The filter bags were put into a 30 mL crucible with known weight (M6), washed at a temperature of 600 °C $\pm$ 15 °C for 2 h, and weighed (M7). Hemicellulose content% = (M3 − M4)/M1 $\times$ 100%; cellulose content% = (M4 − M5)/M1 $\times$ 100%; lignin content% = ((M5 − M1) − (M7 − M6))/M1 $\times$ 100%.

The dried stems and leaves were ground into powder, 0.2 g of sample was weighed into the digestion tube, and 5 mL of concentrated $H_2SO_4$ was added into the same digestion tube. The digestive tubes were put into a heating digester (VELP Scientifica) with a 2-stage increase in temperature to 360 °C, then samples were cooled slightly, and 2 mL of 30% $H_2O_2$ was added into every digestion tube. After 10 min, the operation of adding hydrogen peroxide was repeated several times until the sample was digested completely. The digestion samples were used for the subsequent analysis [32].

A continuous flow analyzer (AA3) was used for nitrogen and phosphorus content analysis of digested samples. The measurement wavelength was 660 nm, the injection time was 60 s, and the flushing time was 12 s [32,33].

The standard curves were made by 0, 10, 20, 30, and 40 mg/L potassium chloride. The digestion samples were measured using a flame photometer, and the potassium content was calculated according to the value on the flame photometer and the standard curve [34].

### 2.4. Phylogenetic Analysis and Transactivation Activity Assay of NtHB8sy, NtHB8to, NtNST3sy, NtNST3to

Multiple sequence alignments of *HB8* and *NST* protein sequences (Supplementary File S1) at the amino acid level were performed using the ClustalW program. A neighbor-joining (NJ) phylogenetic tree was then generated based on the alignment result using MEGA 7.0 with the following parameters: Poisson model, partial deletion, and bootstrap values (1000 replicates).

The method of transactivation activity assay of analysis was described by Na et al. [35]. Simply, the PCR products of *NtHB8sy*, *NtHB8to*, *NtNST3sy*, and *NtNST3to* were combined into the *pBD-GAL4* vector via EcoRI/SalI sites with specific primers, and with the *pBD-GAL4* vector as the negative control. The yeast strain AH109 was used to test transcriptional activation activity. The positive colonies were transferred to the SD medium lacking leucine, tryptophan, histidine, and adenine (QDO, SD/-Leu-Trp-His-Ade) supplemented with X-$\alpha$-Gal plates at 30 °C for 4 days.

### 2.5. Transcription-Polymerase Chain Reaction (qRT-PCR) Analysis

The marked stems of each plant space treatment were collected and immediately frozen in liquid nitrogen and stored at −80 °C for RNA extraction. Three biological replicates were employed per sample. The method of qRT-PCR analysis was described by Na et al. [35]. Because *Nicotiana tabacum* was heterotetraploid with *Nicotiana. sylvestris* (sy) and *Nicotiana tomentosiformis* (to) as subgenome donors, all genes were detected separately according to the subgenomes. The qRT-PCR primers of *NtHB8sy*, *NtHB8to*, *NtNST3sy*, *NtNST3to* are shown in Supplementary File S2.

### 2.6. cis-Acting Regulatory Element Analysis

The genome sequences were used to retrieve the 3kb upstream regions for each gene (Supplementary File S3). *Cis*-acting element analyses of *NtHB8sy*, *NtHB8to*, *NtNST3sy* and *NtNST3to* were carried out using the PlantCARE database (http://bioinformatics.psb.ugent.be/webtools/plantcare/html/, accessed on 4 September 2022) [36].

### 2.7. Statistical Analysis

All statistical analyses were carried out with R software, version 4.2.0. One-way ANOVA Tukey HSD test was used for the comparison between treatments. Pearson's method was engaged to calculate the correlation coefficient.

## 3. Results

### 3.1. Effect of Plant Spacing on Vascular Bundle of Tobacco Stem

From edge to the core, the stem was composed of epidermis, cortex, vascular bundle, and pith (in the center), and the cell type of cortex and pith was parenchyma cell. The vascular bundle of the tobacco stem, which contained phloem, cambial, and xylem, was a bicollateral vascular bundle, i.e., with phloem on both sides of the xylem. (Figure 2A). The phloem fibers are visible clearly in mature vascular tissue (Figure 2B,F). According to the anatomical structure of the tobacco stem cross section, the cell number in the xylem was higher than that in the phloem. It could be seen that cambium cells divide one phloem cell outward and several xylem cells inward. Cambial cells are composed of stem cells and arranged orderly, new phloem cells (phloem mother cell) differentiate on the peripheral side of the cambium, and new xylem cells (xylem mother cell) differentiate on the internal side. The initiation of phloem and xylem from primary cells were identified as showed in Figure 2C. The xylem of the tobacco stem was mainly composed of the primary xylem (Figure 2E) and secondary xylem (Figure 2D), and it contained vascular rays, xylem fiber, and xylem vessel.

The development stage of the tobacco stem treated with different plant spacing was similar on the marking day. On the 4th day after marking, the vascular bundles of tobacco stems under plant spacing of 30 and 40 cm were in the transition stage (Figure 3A). Typical secondary structures of vascular bundles in tobacco stems under plant spacing of 50 and 60 cm were basically formed. It could be seen from Figure 3B that the thickness of the primary xylem under plant spacing of 50 and 60 cm was similar, but the thickness of the secondary xylem under plant spacing of 60 cm was significantly greater than that of 50 cm. On the 12th and 40th day after marking, the xylem thickness under the treatment with plant spacing of 60 cm was significantly greater than that in the other treatments (Figure 3C,D).

With the development of the stem, the phloem thickness of each treatment increased gradually. Forty days after marking, the phloem thickness in 60 cm plant spacing was significantly thicker than in 30 cm (Figure 3E). With the stem thickening, the cambium thickness of each treatment was showing a trend of increasing at first and then decreasing. The cambium thickness in 30 cm plant spacing reached the peak at 4 days after marking. The cambium thickness in 40–60 cm plant spacing reached the peak 12 days after marking. The cell number of cambium in 60 cm plant spacing decreased slightly 40 days after marking. (Figure 3F). Four days after marking, the xylem thickness in the 60 cm plant-spacing treatment was significantly greater than that in the 30 cm treatment, and the xylem

thickness of 40 cm and 50 cm plant-spacing treatment was significantly greater than that in the 30 cm treatment 12 days after marking. Forty days after marking, there was no significant difference in xylem thickness under the different plant spacing. However, the xylem thickness in 30–50 cm plant spacing was significantly less than that in the 60 cm treatment (Figure 3G). Four to twelve days after marking, the maximum vessel diameter in the 60 cm treatment was significantly larger than in 30 cm plant spacing, and there was no significant difference in the maximum vessel diameter among all treatments 40 days after marking (Figure 3H). The results showed that plant spacing could affect vascular bundle thickness and then stem girth by regulating cambium cell activity.

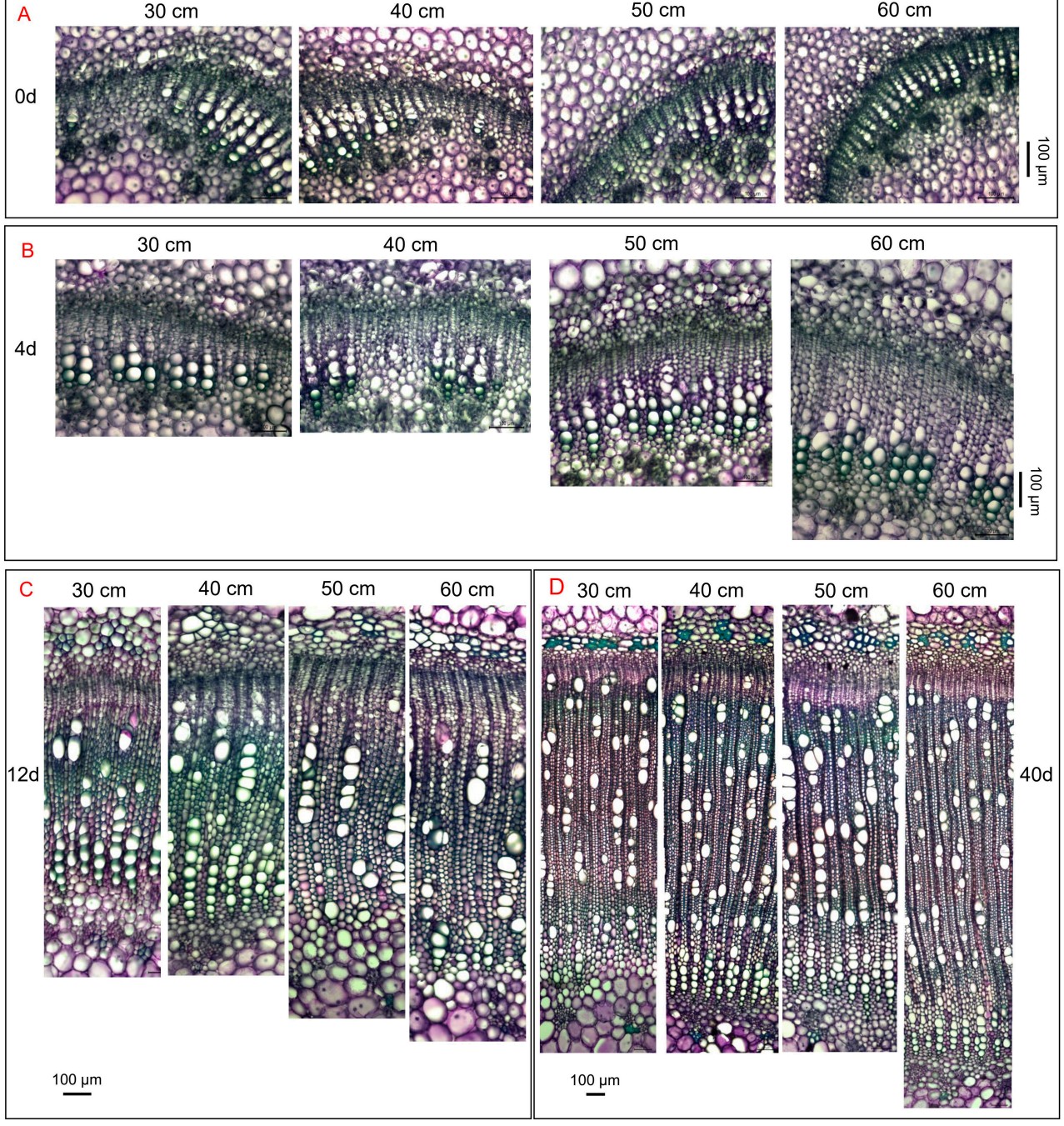

**Figure 3.** *Cont.*

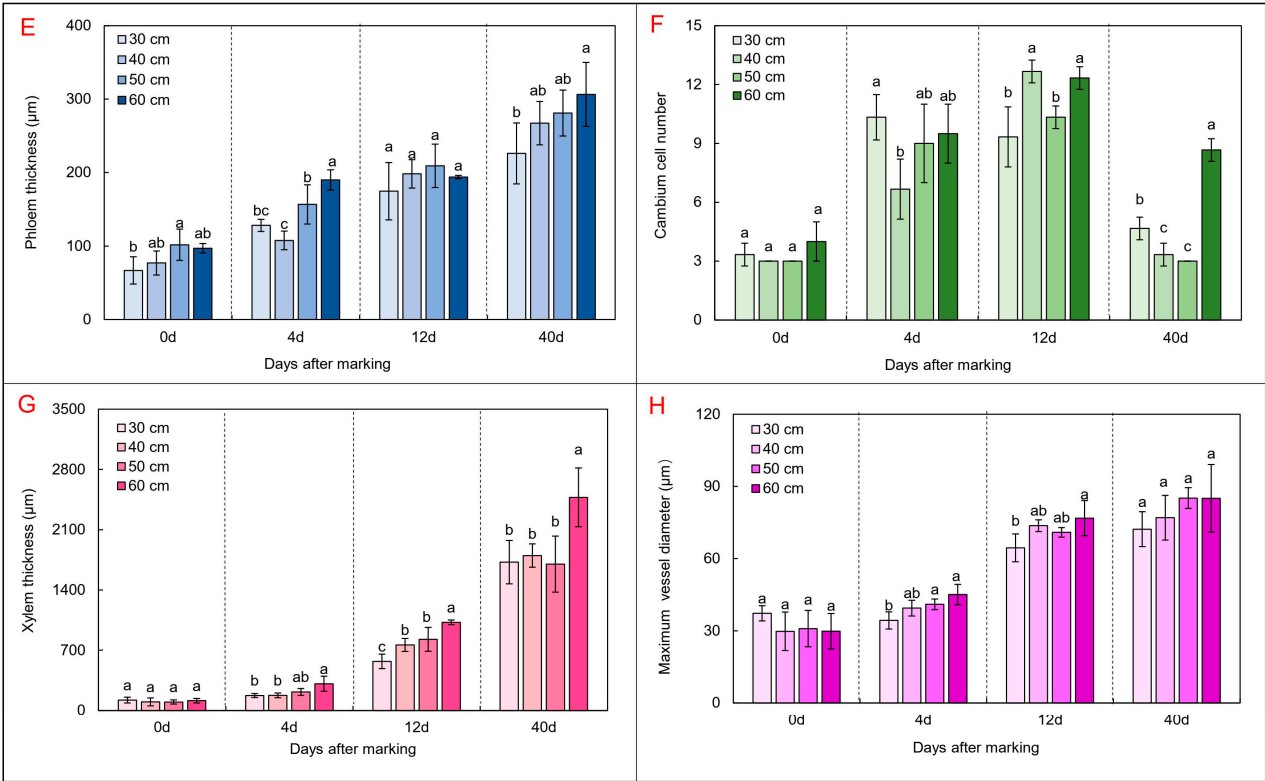

**Figure 3.** Cross section of the vascular bundle of tobacco stem under different plant-spacing treatments at 0 d, 4 d, 12 d, and 40 d (**A–D**). The scale bar of every section is 100 μm. The key indexes of the vascular bundle are shown in (**E–H**). The bars represent the standard error of the mean, and the letters above each bar represent significant differences between different treatments ($p < 0.5$).

### 3.2. Effect of Plant Spacing on Secondary Cell Wall Formation

Except for the secondary cell wall thickness of xylem vessels in 50 and 60 cm plant spacing being significantly greater than that in 30 cm on the day of marking, there was no significant difference in the secondary cell wall thickness of xylem vessels in other periods (Figure 4A). The secondary cell wall thickness of wood fiber in 30 cm plant spacing was significantly greater than that in 50 cm on the 12th day after marking, and there was no significant difference in the thickness of the secondary cell wall among all treatments on the 40th day after marking (Figure 4B). On the 12th day after marking, the secondary cell wall thickness of phloem fibers in 60 cm plant spacing was significantly greater than that in the other treatments. On the 40th day after marking, the secondary cell wall thickness of phloem fibers in 30 cm and 40 cm plant spacing was significantly greater than that in 50 and 60 cm treatments (Figure 4C). The thickness of the secondary cell wall of the perimedullary phloem fibers in 30 cm and 50 cm plant spacing was significantly greater than that in the 40 cm and 60 cm treatments on the 12th day after marking, and the thickness of the secondary cell wall of the perimedullary phloem fibers in 60 cm plant spacing on the 40th day after marking was significantly greater than that in the other treatments (Figure 4D). Forty days after marking, the lignin content in 60 cm plant spacing was significantly higher than that in 30 cm spacing, and the lignin content in 30 cm and 60 cm plant spacing was significantly higher than in 40 cm and 50 cm spacing. The cellulose content in 60 cm plant spacing was significantly higher than in the other treatments, and there was no significant difference between the hemicellulose content among the treatments. (Figure 4E).

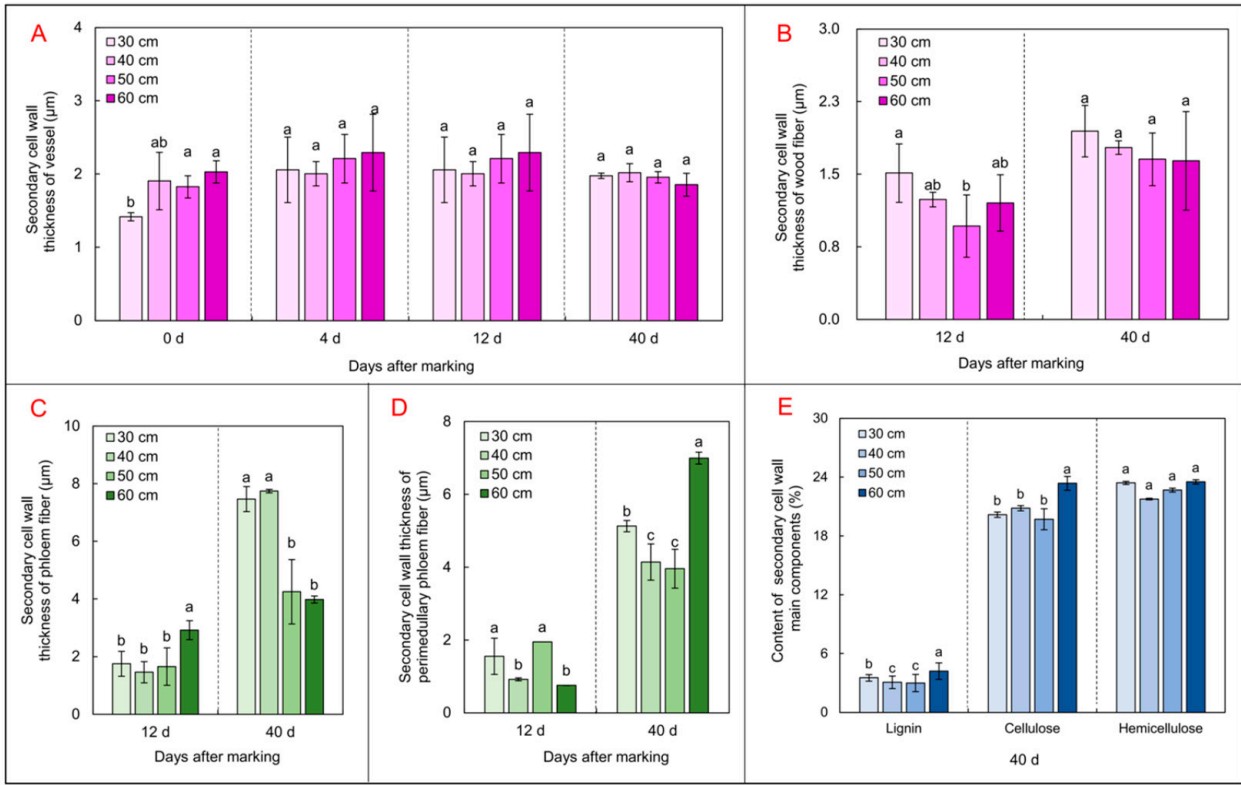

**Figure 4.** Effect of plant spacing on secondary cell wall thickness of xylem and phloem cells (**A**–**D**). Effect of plant spacing on the content of secondary cell wall main components (**E**). The bars represent the standard error of the mean, and the letters above each bar represent significant differences in different treatments ($p < 0.5$).

### 3.3. Effect of Plant Spacing on NtHB8s and NtNST3s Genes Expression

Different plant spacing leads to variation in light transmittance in plant cultivation. The light transmittance between plants and between rows was measured. The result is demonstrated in Figure 5A: the light transmittance increased along with plant spacing, but there is no difference between the plant spacing of 50 and 60 cm.

The full lengths of *NtHB8sy*, *NtHB8to*, *NtNST3sy*, and *NtNST3to* were cloned and inserted into the *pBD-GAL4* vector. It could be seen from Figure 5B that the yeast transformant constructed with *pBD-GAL4-NtHB8sy*, *pBD-GAL4-NtHB8to*, *pBD-GAL4-NtNST3sy*, and *pBD-GAL4- NtNST3to* turned blue in the presence of X-α-Gal and grew normally on the SD medium (SD/-Leu-Trp-His-Ade), whereas the negative control did not. *NtHB8sy* and *NtHB8to* are classed into the same group and are homologous to *AtHB8*. *NtNST3sy* and *NtNST3to are* classed into the same group and are homologous to *AtNST3* (Figure 5C). The expression patterns of *NtHB8sy*, *NtHB8to*, *NtNST3sy*, and *NtNST3to* on the day of marking were identical; gene expression levels in 40 cm plant spacing were significantly higher than those in 30 cm and 50 cm, and 30 cm and 50 cm levels were significantly higher than that in 60 cm. Four days after marking, the gene expression levels of *NtHB8sy* and *NtNST3to* in the 40 cm treatment were significantly higher, and the expression of *NtHB8to* and *NtNST3sy* in the 30 cm treatment was significantly higher than in the other treatments. The gene expression levels of *NtHB8sy*, *NtHB8to*, and *NtNST3sy* in 60 cm treatment were significantly higher, and the gene expression level of *NtNST3to* in the 40 cm treatment was significantly higher than in the other treatments 12 days after marking. The regularity of each gene expression level was that the earlier of each gene expression reached the peak in the smaller of the plant spacing (Figure 5D).

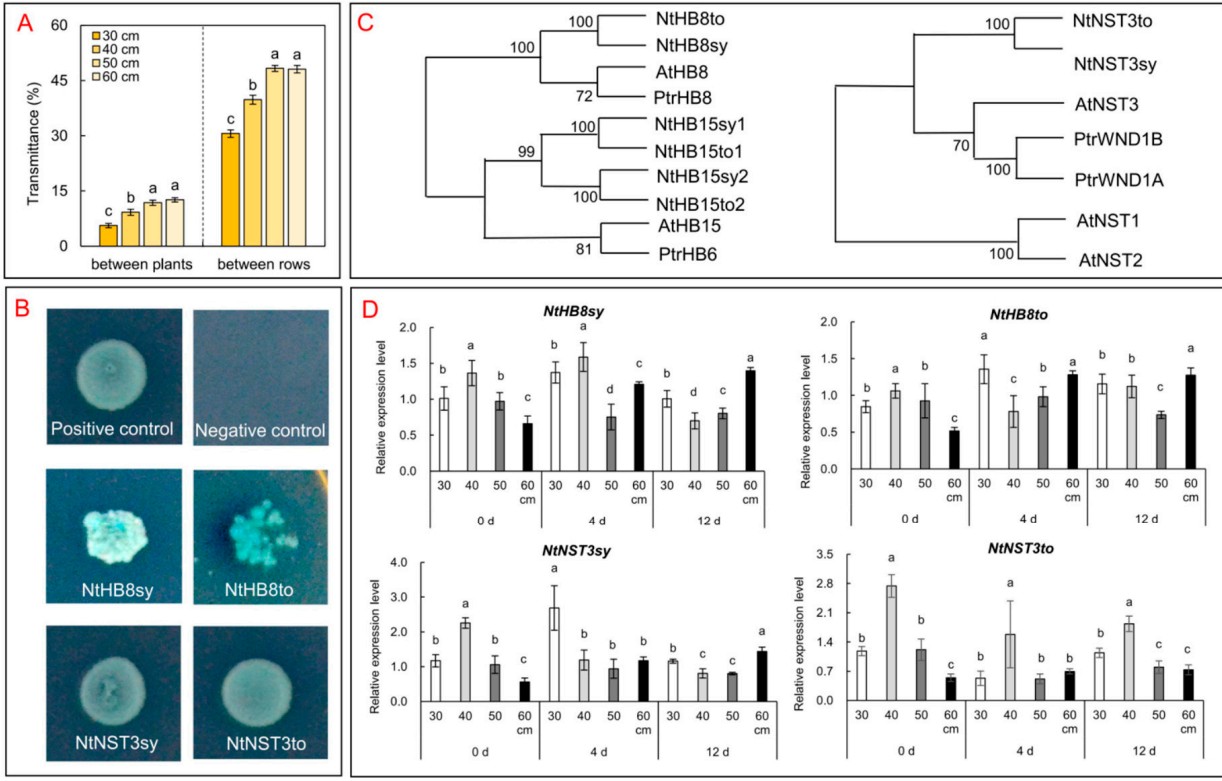

**Figure 5.** (**A**) The light transmittance between the plants and between rows was measured. (**B**) Transactivation analysis of *NtHB8sy*, *NtHB8to*, *NtNST3sy* and *NtNST3to*. Genes were combined into the GAL4 (BD) DNA binding domain in pBD-GAL4, with the gene having transcriptional activation activity as the positive control, and empty pBD-GAL4 vector as the negative control. (**C**) Neighbor-joining (NJ) phylogenetic relationships of *HB8s* and *NSTs*. (**D**) qRT-PCR analysis of *NtHB8sy*, *NtHB8to*, *NtNST3sy*, and *NtNST3to* expression of tobacco stem under 30, 40, 50, and 60 cm plant-spacing treatments after tobacco stem marking. The bars represent the standard error of the mean, and the letters above each bar represent significant differences in different treatments ($p < 0.5$).

### 3.4. cis-Acting Element Analysis of Key Genes

The 3 kb upstream regions of *NtHB8* and *NtNST3* were analyzed in the PlantCARE database to find useful information about the regulatory mechanism (Table 1). For promoters studied, all of them contained light-response elements, hormone-responsive elements, low-temperature elements, MYB, and MYC elements. Abundant light-response elements indicated that *NtHB8* and *NtNST3* were related to light responsiveness. *NtHB8* and *NtNST3* might relate to hormone and stress responsiveness, as well as be regulated by MYB and MYC transcription factors. However, the elements of allelic gene promoters were not identical, such as only *NtHB8sy* had anaerobic elements, only *NtHB8to* had SA elements, and only *NtNST3sy* had ABA elements. The number of the same elements in allelic gene promoters was not identical either.

**Table 1.** The distribution of main cis-acting elements in the 3 kb upstream promoter regions of key genes.

| Gene | Light | Circadian | Auxin | ABA | GA | MeJA | SA | Anaerobic | Low-Temperature | MYB | MYC |
|------|-------|-----------|-------|-----|-----|------|-----|-----------|-----------------|-----|-----|
| *NtHB8syP* | 18 | 1 | / | 1 | / | 2 | / | 4 | 3 | 14 | 8 |
| *NtHB8toP* | 19 | 1 | / | 3 | / | 4 | 2 | / | 2 | 9 | 8 |
| *NtNST3syP* | 20 | / | 2 | 6 | 2 | 6 | 4 | / | 2 | 8 | 2 |
| *NtNST3toP* | 12 | / | 2 | / | 2 | 10 | 1 | / | 1 | 4 | 7 |

### 3.5. Effect of Plant Spacing on Nutrient Content

The total accumulation of nitrogen, phosphorus, and potassium in leaves and stems of tobacco in 60 cm plant spacing was significantly higher than in the other treatments, and there was no significant difference among the treatments (Figure 6A). The accumulation of nitrogen, phosphorus, and potassium in leaves in each treatment was higher than that in stems of corresponding parts. The nitrogen content in leaves in 60 cm plant spacing was significantly higher than that in the other treatments, and there was no significant difference among these treatments. Nitrogen content in stems in 40, 50, and 60 cm plant spacing was significantly higher than that in 30 cm plant spacing (Figure 6B). The phosphorus content in leaves in 60 cm plant spacing was significantly higher than that in 30 and 40 cm plant spacing. The accumulation of phosphorus in stems in 60 cm plant spacing was significantly greater than in 30 and 40 cm plant spacing (Figure 6C). The accumulation of potassium in leaves in 60 cm plant spacing was significantly higher than in these treatments and lower than that in 40 cm plant spacing (Figure 6D).

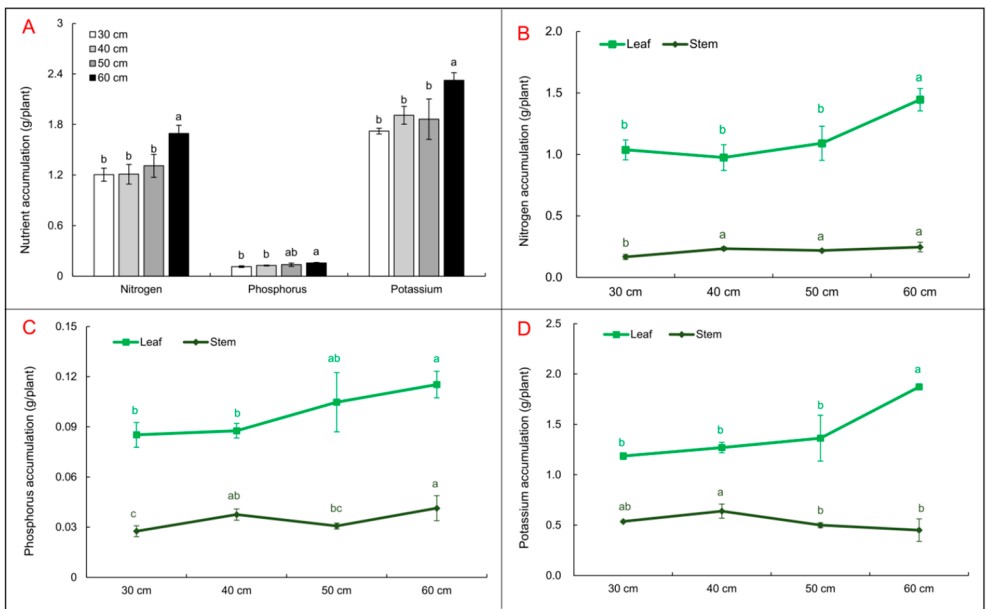

**Figure 6.** Total accumulation of nitrogen, phosphorus, and potassium in marked stems and leaves (**A**). The accumulation of nitrogen (**B**), phosphorus (**C**), and potassium (**D**) in marked stems and leaves. The bars represent the standard error of the mean, and the letters above each bar represent significant differences in different treatments ($p < 0.5$).

## 4. Discussion

### 4.1. Anatomic Structure and Gene Expression Pattern Analysis of the Transition from Primary to Secondary Stem Development

Secondary growth usually occurs in gymnosperms and dicotyledons [37]. Poplar and Arabidopsis have proven to be an excellent model for studying secondary growth [9]. In cross section, the stem of poplar is similar to the hypocotyl of Arabidopsis, such as both plants have vessel elements, sieve-tube elements, companion cells, and parenchyma cells and fibers [38]. However, rays, radially organized files of parenchyma cells in the secondary vascular tissues of hybrid aspen, were not unequivocally identified in Arabidopsis [39]. Although Busse and Evert observed rays in the secondary xylem of Arabidopsis hypocotyls (under proper growth conditions), they are obviously not the common feature of secondary growth elongation in Arabidopsis [40]. In addition, secondary growth is only to be produced at the basal region of the Arabidopsis shoot [4], and the regulatory network in poplar is more complex [41]. The anatomical structures of the tobacco stem include the cortex, phloem, cambial, xylem, perimedullary phloem, and pith. The vascular rays, xylem fiber, and xylem vessel in the xylem of tobacco are similar to those in poplar (Figure 2). However,

there is also some difference between tobacco and poplar, such as the pith in tobacco is obvious (Figure S1) and the tobacco stem has perimedullary phloem.

The secondary structure was formed completely 8 days after marking. The maximum increase in xylem thickness, the maximum xylem vessel diameter, and the size of pith cells occurred 8 to 12 days after marking. The increase in stem girth was mainly due to the increase in cell number and cell diameter. Cells of the vascular bundle were derived from the cambium, and the cambium cell number in the active cambium was higher than that in the dormant cambium [9,42]. The number of cambium cells increased 4 to 12 days after marking, which indicated that the cambium was in an active state at this stage [43]. The cambium-divided cells differentiate into the xylem, resulting in vascular bundle thickening, which leads to the thickening of stem girth. Twelve days after marking, the number of cambium cells began to decline, which indicated that the ability of cambium division was gradually falling, which was also the main reason for the decrease in stem girth growth (Figures 3 and S2).

Development of the xylem and phloem involves cell division, cell expansion, formation of secondary cell walls (involving cellulose, hemicellulose, and lignin synthesis), and programmed cell death [44]. During the secondary growth of poplar, HB and NAC gene family members show strong secondary growth-associated upregulation [45]. Lu et al. reported the anatomical structure of the xylem and secondary cell wall when tobacco was used as transgenic receptor material of the poplar gene [46]. To date, there is no systematic report on the secondary growth of tobacco stems. On the other hand, there was no report about the key genes associated with secondary growth, for instance, *ATHB-8* homologous gene in tobacco. In the present research, *NtHB8sy*, *NtHB8to*, *NtNST3sy*, and *NtNST3to*, homologous with corresponding genes in Arabidopsis and poplar, were cloned; we also proved that all of them had transcriptional activation activity, which indicated that they were function genes. As for the secondary growth of tobacco, the expression levels of *NtHB8sy*, *NtHB8to*, *NtNST3sy*, and *NtNST3to* kept varying, which was in affirmation with the earlier findings and implied their role in vascular differentiation (Figure 5).

*4.2. Morphological Structure, Chemical Composition, and Gene Expression Pattern Response of Tobacco to Plant Spacing*

The development stage of the tobacco stem under different plant-spacing treatments was basically the same on the day of marking because the size of the tobacco was small and their development was not limited. With the thickening of the stem, the thickness of the cambium under every treatment tended to increase first and then decrease. The thickness of the cambium under the 30 cm plant-spacing treatment reached the peak on the 4th day after marking, and the thickness of the cambium in the other treatments reached the peak on the 12th day after marking (Figure 3F). It can be seen that the decrease in plant spacing could impact the activity of the cambium—the smaller the plant spacing, the earlier the activity of the cambium decreases. The effects of plant spacing on cambium activity and vascular bundle development were consistent. The greater the plant spacing, the higher the cambium activity, and the greater the thickness of the xylem and phloem. These findings are found to be similar to maize [47].

The vascular system fulfills two main functions, long-distance transport and mechanical support. Xylem cells, with thick secondary cell walls rich in lignin, cellulose, and hemicellulose, play an important role in providing support to the plant and transporting water, nutrients, and minerals from the root to the shoot [48]. The lignin content in 30 and 60 cm plant spacing on the 40th day after marking was significantly higher than that in the other treatments, but the reasons for the higher lignin content in these two treatments might be different (Figure 4E). The thickness of the xylem in 60 cm plant spacing and the thickness of the secondary cell wall of phloem fiber was significantly greater than that in the other treatments. The lignin content in the 60 cm plant-spacing treatment was significantly greater than that in the other treatments, which might be caused by the large number of lignified cells and the large thickness of the secondary cell wall of phloem fiber. The high

lignin content in 30 cm plant spacing might be caused by too small plant spacing, which limited the development of tobacco plants and promoted the lignification of tobacco plants. The gene expression patterns of *NtHB8sy*, *NtHB8to*, *NtNST3sy*, and *NtNST3to* on the 12th day after marking were also sufficient proof of the above inference (Figure 5D).

The light might be playing an important role in the procedure depicted above. As demonstrated in the result part, the transmittance was influenced largely by plant spacing. Simultaneously, the 3 kp upstream regions of these genes were abundant with the motifs banding light-responding transcription factors. These results implied that the expression pattern of these genes could be determined by different light conditions due to the plant spacing variation.

On the other hand, plant spacing could also lead to other subjective conditions for an individual crop, for instance, delivery of water, essential mineral nutrients, sugars, and amino acids; transportation of those nutrients to the various plant organs is the essential function performed by the vascular system [49]. To date, the effect of plant spacing on the chemistry and physiology of plants is extremely understudied [50]. In this study, the accumulation of nitrogen, phosphorus, and potassium in leaves and stems increased with the increase in plant spacing. The accumulation of nitrogen, phosphorus, and potassium in stems and leaves in the 60 cm plant-spacing treatment was significantly greater than that in 30 cm plant spacing. Close planting might reduce the nutrients in the soil corresponding to a single plant, and then affect the aboveground nutrient accumulation. It could be seen that the increase in plant spacing is helpful to the transportation and accumulation of nutrients from soil to plant. Interestingly, with the increase in plant spacing, the potassium accumulation in stems decreased significantly, while the potassium accumulation in leaves increased significantly. The sum of potassium accumulation in stems and leaves increased significantly with the increase in plant spacing. Therefore, the increase in plant spacing contributed to the accumulation of potassium and significantly promoted the transport of potassium to leaves (Figure 6).

Close planting will lead to the shading of leaves among plants and affect the photosynthetic photo flux density of plants. The present research showed that the light transmittance between plants and rows was highly correlated with the plant-spacing treatments (with correlation coefficients 0.97 and 0.94 of light transmittance between plants and rows, respectively), as shown in Figure 7. Cambial cell divisions were controlled by photoperiod [43]. The light could promote xylem fiber-like cellular differentiation and regulate the synthesis of the main chemical components of the secondary cell wall [51,52]. In the present research, we demonstrated that the crop with 60 cm plant spacing at 40 days after marking had the highest cambium cell number; phloem thickness and xylem thickness had a high correlation with the plant spacing at 40 days and 12 days after marking, respectively (Figure 3, both correlation coefficients as high as 0.98); the crop with 60 cm plant spacing had the highest lignin and cellulose deposition in its stem secondary cell walls (Figure 4). Furthermore, the expression of *NtHB8s* and *NtNST3s* in crop stems with different plant spacing also showed varied patterns, and their promoter regions were rich in a *cis*-acting element, which could be bound by light-relevant transacting elements. It implied that plant spacing might affect the expression of *NtHB8s* and *NtNST3s* because of the availability of light and then affect the morphological structure and chemical content of the stem. Close planting will also lead to fewer nutrients absorbed by a single plant, which will affect the stem phenotype [25,53,54]. In this study, the major nutrient (nitrogen, phosphorous, and potassium) deposition in plant stem and leaves also showed a high correlation with plant spacing. Studies have shown that shading has a greater impact on plants than nutrient depletion [55,56]. The hypothesis model of how plant spacing influences tobacco crops in the field in the present is demonstrated in Figure 7.

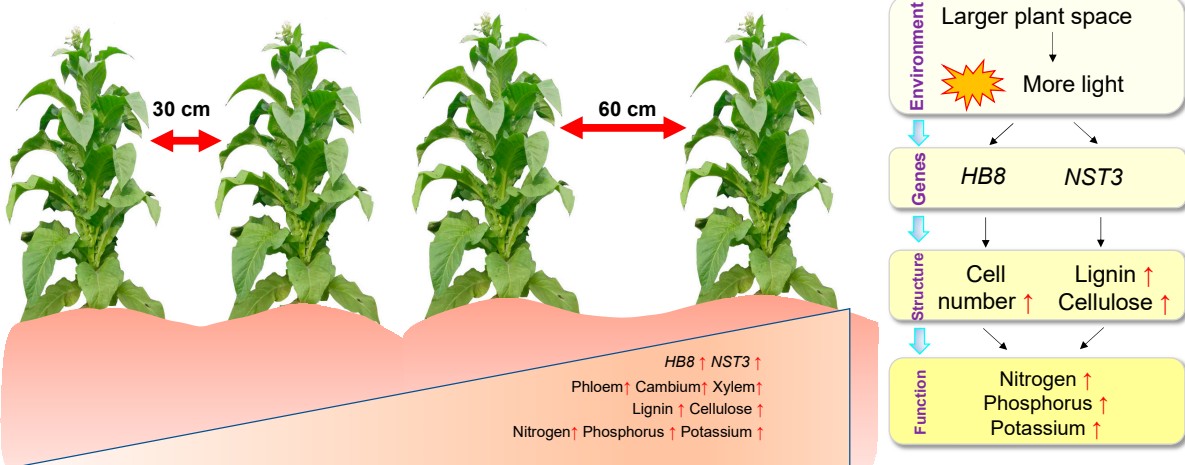

**Figure 7.** The overview of the relationship of "environment-gene-structure-function". Red upward arrows indicate an increase.

## 5. Conclusions

The relationship between plant spacing and crop yield is complicated and affected by many elements. Present research demonstrated that plant spacing in the field could influence crop stem cambium secondary growth. The result shows that higher plant spacing provides field plants with more available light, which could lead to more phloem and xylem in a thicker stem. The phenomenon implied that the stem strength and even the yield per plant might benefit from higher plant spacing. Hopefully, these results could serve as a reference for further studies on relevant aspects.

**Supplementary Materials:** The following supporting information can be downloaded at: https://www.mdpi.com/article/10.3390/agronomy13082142/s1.

**Author Contributions:** W.Q. and Y.Z. conceived and designed the experiments; N.X. and L.M. performed the experiments; F.T., S.D., Y.X., S.K. and Y.L. (Yang Li) participated in data collection and analysis; N.X. wrote the manuscript; W.S. and Y.L. (Yuanda Lv) revised the manuscript. All authors have read and agreed to the published version of the manuscript.

**Funding:** This work was financially supported by the Agricultural Science and Technology Innovation Program (ASTIP-TRIC03), China National Tobacco Corporation Program (110202103014).

**Data Availability Statement:** The data presented in this study are available on request from the corresponding author.

**Acknowledgments:** We thank Yi Shi for guiding the design of the field experiment. We thank Luxin Kong for her help in the secondary wall content analysis.

**Conflicts of Interest:** The authors declare no conflict of interest.

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
