# Peer review of "Plant Spacing Effects on Stem Development and Secondary Growth in Nicotiana tabacum"

_agronomy, doi:10.3390/agronomy13082142_

Round 1

Reviewer 1 Report

 * Looking at the title of the article, it is understood more as an
    agromorphological study. I think the title of the article should be
    changed to emphasize the importance of the article.
  * Line1-2 “/Nicotiana tabacum/ “ should be written italic.
  * Most of the keywords are also included in the abstract. It is
    recommended that keywords should not be used mainly in the abstract
    and title.
  * When we look at the general writing order of the article, it will be
    more understandable to write it as introduction, material and
    method, findings, discussion and conclusion, respectively.
  * Lines 69-77 should be given as a separate paragraph and the purpose
    of the study should be clearly emphasized.
  * Line 115 delete .
  * again line 115 "ZC208" as a tobacco variety or line. Please explain.
  * Line 409-410 “Nicotiana. tabacum was heterotetraploid with
    Nicotiana. sylvestris (sy) as fe-409 male parent and Nicotiana.
    Tomentosiformis”  delete . after Nicotiana. Please check all the text.
  * Based on what you wrote in 409-411, you can give information about
    the parents of the variety or type used in the article.
  * Although there are many places about light, there is no data about
    light. It is not appropriate to comment on a subject without data.
    The effects of planting frequency on light should be examined and
    these should be added to the article.
  * LÄ°NE 314-315 it is mentioned that /"Samples used for stem girth,
    stem anatomical structure, physiological index and gene expression
    analysis were sampled 0, 4, 8, 12, 16, 20, 40 days after marking."/
    although the data is given as only 0,4,12,40 in the tables. these
    need to be corrected. In addition, P values related to these
    parameters should be given. Just lettering is not enough.
  * Numerical data cannot be understood in all charts. must be
    rearranged in a readable manner.
  * Correlation analyzes should be done in order to make comments about
    the connections between the parameters, therefore, these analyzes
    should be added to the study. 

Author Response

Dear respect reviewer

 Our point-to-point response to your kind comments and suggestions is listed below.

 * Looking at the title of the article, it is understood more as an
    agromorphological study. I think the title of the article should be
    changed to emphasize the importance of the article.

Response: Thank you for the comments, we have the title changed into: Plant Spacing Effects on Stem development and Secondary Growth in Nicotiana tabacum.
  * Line1-2 “/Nicotiana tabacum/ “ should be written italic.

Response: Thank you for the comment, we did so in the revised manuscript.

  * Most of the keywords are also included in the abstract. It is
    recommended that keywords should not be used mainly in the abstract
    and title.

Response: Thank you for the comment, we did so in the revised manuscript.
  * When we look at the general writing order of the article, it will be
    more understandable to write it as introduction, material and
    method, findings, discussion and conclusion, respectively.

Response: Thank you for the comment, we did so in the revised manuscript.
  * Lines 69-77 should be given as a separate paragraph and the purpose
    of the study should be clearly emphasized.

Response: Thank you for the comment, we did so in the revised manuscript.
  * Line 115 delete .

Response: Thank you for the comment, we did so in the revised manuscript.
  * again line 115 "ZC208" as a tobacco variety or line. Please explain.

Response: Thank you for the comment, explanation has been made in the new version.
  * Line 409-410 “Nicotiana. tabacum was heterotetraploid with
    Nicotiana. sylvestris (sy) as fe-409 male parent and Nicotiana.
    Tomentosiformis”  delete . after Nicotiana. Please check all the text.
  * Based on what you wrote in 409-411, you can give information about
    the parents of the variety or type used in the article.

Response: Thank you for the comments, after checking this part, we found it’s a problematic statement, which has been thus revised in the new version.
  * Although there are many places about light, there is no data about
    light. It is not appropriate to comment on a subject without data.
    The effects of planting frequency on light should be examined and
    these should be added to the article.

Response: Thank you for the comments, we have had the light transmittance was measured and the data is demonstrated in Figure 5A.  
  * LÄ°NE 314-315 it is mentioned that /"Samples used for stem girth,
    stem anatomical structure, physiological index, and gene expression
    analysis were sampled 0, 4, 8, 12, 16, 20, 40 days after marking."/
    although the data is given as only 0,4,12,40 in the tables. these
    need to be corrected. In addition, P values related to these
    parameters should be given. Just lettering is not enough.

Response: Thank you for the comments, it is a mistake during manuscript writing, we apologize for it, and in the new version it has been corrected. There is a lot of p value calculated to determine if the measured parameters between treatments were significantly different or not by ANOVA. So in the column charts, we used different letter to mark the significant difference (p<0.05).
  * Numerical data cannot be understood in all charts. must be
    rearranged in a readable manner.

Response: Thank you for the comments, chart pictures with much higher resolution have been used.
  * Correlation analyses should be done in order to make comments about
    the connections between the parameters, therefore, these analyzes
    should be added to the study. 

Response: Thank you for the comments, in the new version correlation analysis has been done to emphasize the relationship between the varied crop traits and the plant spacing treatments. The relationship is discussed mainly in the last paragraph of the discussion.

Hopefully, we could address all your concern. Thanks again for your effort and attention.

Best wishes

Authors

Reviewer 2 Report

1. Manuscript title: It should be more specific to point out the main theme, particularly about the expression of two key transcription factors.

2. I don't understand what you mean by "tissue transmission" and "stem secondary growth dynamics", so please define them more clearly.

3. L23-24, L27-29: They should be rewritten to make them more fluent.

4. Keywords: They are key terms but did not appear in the manuscript title.

5. The authors need to provide some iconic photos of the treated and the control plants to present their overall performance in the field trials.

6. In this study only two genes were cloned and analyzed, so the suggestion is to test more genes related to light response to validate their hypothesis.

7. The authors claim that more space can get more light in the field, but they did not provide data for this. What is the difference in total light exposure between treatments?

8. The conclusion should be improved together with the legend of Figure 7. The authors need to give perspective on the future direction of studying the relationship between plant spacing, light quantity and light response of plants, etc.

9. The resolution of all figures is very low and thus it's very difficult to analysis all data together with the significant effects between means.

Author Response

Dear respect reviewer

 Our point-to-point response to your kind comments and suggestions is listed below.

  1. Manuscript title: It should be more specific to point out the main theme, particularly about the expression of two key transcription factors.

Response: Thank you for the comments, we have the title changed into: Plant Spacing Effects on Stem development and Secondary Growth in Nicotiana tabacum.

  1. I don't understand what you mean by "tissue transmission" and "stem secondary growth dynamics", so please define them more clearly.

Response: Thank you for the comments, we have the explain these two terms in the introduction part of the new manuscript line 40-45.

  1. L23-24, L27-29: They should be rewritten to make them more fluent.

Response: Thank you for the comment, we did revision in this part to improve it.

  1. Keywords: They are key terms but did not appear in the manuscript title.

Response: Thank you for the comment, please check the new article title.

  1. The authors need to provide some iconic photos of the treated and the control plants to present their overall performance in the field trials.

Response: Thank you for the comment, the tobacco plant in the field is quite large which is difficult for photographing. On the other hand, in the manuscript, we have provided iconic anatomic pictures for instance in Figure 3 A - D. 

  1. In this study only two genes were cloned and analyzed, so the suggestion is to test more genes related to light response to validate their hypothesis.

Response: Thank you for the comments, tetraploid tabaco is a plant species with less genomic and genetic information than the other crops for instance rice, maize, or the model plant Arabidopsis. On the other hand, it is quite important because of its potential in medicine use and the model plant diploid tabaco which is its close relative species. NB8 and NST3 are well-known transcriptomic factors essential in plant stem development and secondary development. However, there is no information about their homologous in tabaco reported ever.  So here we clone the NtNB8s and NtNST3s and tested their transcription activation ability. Here we also provided the deduced promoter region sequences and showed the possible binding transcription factor. Furthermore, we also demonstrated the expression level dynamic of these coding genes. All the data mentioned above implied NtNB8s and NTNST3s could be involved and play a role in the molecular regulation responding to the different plant spacing treatments. Of course, it is very good suggesting to study other genes which might be involved in this procedure, and our new research will be carried out focusing on this point.        

  1. The authors claim that more space can get more light in the field, but they did not provide data for this. What is the difference in total light exposure between treatments?

Response: Thank you for the comments, we have had the light transmittance measured and the data is demonstrated in Figure 5A. 

  1. The conclusion should be improved together with the legend of Figure 7. The authors need to give perspective on the future direction of studying the relationship between plant spacing, light quantity and light response of plants, etc.

Response: Thank you for the comment, the reference to Figure 7 has been removed from the conclusion part. The conclusion part has been revised as well according to the advice.

  1. The resolution of all figures is very low and thus it's very difficult to analyze all data together with the significant effects between means.

Response: Thank you for the comments, chart pictures with much higher resolution have been used.

Hopefully, it could address all your concern. We want to thank again for you kind work on our manuscript.

Best wishes

Authors

Reviewer 3 Report

The manuscript discusses the effect of a different tobacco growing pattern on plant architecture and stem composition. In a way, this is a well-known issue, because the distance between plants within a stand has a fundamental effect on a whole range of indicators, including yield. The authors also supplement the study with an analysis of the effect of spacing on selected genes expression.

At the end of the introduction chapter, defined work hypotheses and work objectives are missing.

Chapter 4 - Material and Methods should be moved before the Results chapter.

Page 11 - line 335 - add full latin name and remove dot. Same page - correct line 341.

The Material and Methods section lacks information about when and where the experiment was performed.

Information on statistical data processing is missing.

Figure 7 should be placed in the text and not after the conclusions.

A reference to the figure should not be given in the conclusions section - the conclusion should only be generalizing.

The cited literature is adequate to the topic and includes over 50 relevant sources.

In principle, the submitted study is fine, and after making adjustments, I would recommend accepting the contribution for publication.

Examples of minor typos are marked in the attached pdf.

From a linguistic point of view, everything appears to be in order, however, I recommend a careful spelling check, as it is possible to find minor errors in the manuscript.

Author Response

Dear reviewer

Here we submit our point-to-point response to your kind suggestions and comments. They are listed below.

Chapter 4 - Material and Methods should be moved before the Results chapter.

Response: Thank you for the comment, we did so in the revised manuscript.

Page 11 - line 335 - add full latin name and remove dot. Same page - correct line 341.

Response: Thank you for the comment, we did so in the revised manuscript.

The Material and Methods section lacks information about when and where the experiment was performed.

Response: Thank you for the comment, we added the relevant information into the new-version manuscript.
Information on statistical data processing is missing.

Response: Thank you for the comment, we added a subsection showing how the statistics had been done into the new-version manuscript.

Figure 7 should be placed in the text and not after the conclusions.

A reference to the figure should not be given in the conclusions section - the conclusion should only be generalizing.

Response: Thank you for the comment, the reference to Figure 7 has been removed from the conclusion part.

Hopefully, our response could address all your concerns, and we want to thank you again for your kind input in our manuscript.

Best wishes

Authors

Round 2

Reviewer 1 Report

Most of the corrections I requested have been made. However, it does not appear in the article and tables as it is written between lines 104-107. I mentioned it in my previous report and although it was written that it was corrected, no correction was made on this issue.

"The stem circumference of the marked nodes 104 was measured every 4 days. Samples used for stem girth, stem anatomic structure, physiological index and gene expression analysis were sampled 0, 4, 12, 16, 40 days after marking. "

The results of gene expression analyzes are given in the range of 0-12 days in some tables and 0-20 days in others. These parts are corrected in accordance with the material method.

Author Response

Most of the corrections I requested have been made. However, it does not appear in the article and tables as it is written between lines 104-107. I mentioned it in my previous report and although it was written that it was corrected, no correction was made on this issue.

"The stem circumference of the marked nodes 104 was measured every 4 days. Samples used for stem girth, stem anatomic structure, physiological index and gene expression analysis were sampled 0, 4, 12, 16, 40 days after marking. "

Response: Thank you for the comments, this part has been corrected and highlighted in the context.

The results of gene expression analyzes are given in the range of 0-12 days in some tables and 0-20 days in others. These parts are corrected in accordance with the material method.

Response: Thank you for the comments, the expression level analysis showed in Figure 5 are given ranged 0-12 day, but in Figure s3, 1-20 day. In the new version however, Figure s3 has been removed, in case of misunderstanding. The relevant context has also been revised accordingly and highlighted.

Reviewer 2 Report

The manuscript has been improved and I don't have further questions.

Author Response

Thank you very much!

Reviewer 3 Report

The authors incorporated all my comments into the revised manuscript.

They also added their comments to individual comments.

All changes are in the updated manuscript.

From my point of view, I recommend accepting the manuscript for publication in its current form.

Author Response

Thank you very much!

Round 3

Reviewer 1 Report

Desired corrections have been made. The paper can be published in this form.